# Potential Use of Deep-Sea Sediment Bacteria for Oil Spill Biodegradation: A Laboratory Simulation

**DOI:** 10.3390/microorganisms10081616

**Published:** 2022-08-10

**Authors:** Tri Prartono, Angga Dwinovantyo, Syafrizal Syafrizal, Agung Dhamar Syakti

**Affiliations:** 1Department of Marine Science and Technology, Faculty of Fisheries and Marine Sciences, IPB University, Jl. Agatis IPB Dramaga Campus, Bogor 16680, Indonesia; 2Research Center for Oceanography, The National Research and Innovation Agency (BRIN) of the Republic of Indonesia, Jl. Pasir Putih Raya No. 1, Jakarta 14430, Indonesia; 3Laboratory of Chemical Properties of Oil and Product Test, Research and Development Centre for Oil and Gas Technology (LEMIGAS), Ministry of Energy and Mineral Resources of the Republic of Indonesia, Jl. Ciledug Raya Kavling 109, Jakarta 12230, Indonesia; 4Marine Sciences Department, Marine Science and Fisheries Faculty, Raja Ali Haji Maritime University, Jl. Politeknik Senggarang, Tanjungpinang 29100, Indonesia

**Keywords:** bioremediation, deep-sea sedimentary bacteria, GC-MS, microcosm, oil degradation

## Abstract

Deep-sea sedimentary hydrocarbonoclastic bacteria are still not widely used in the bioremediation field, especially for crude oil spill biodegradation. This study utilized a mixed culture of *Raoultella* sp., *Enterobacter* sp., and *Pseudomonas* sp. isolated from deep-sea sediment to determine the abilities of bacteria to degrade petroleum hydrocarbons while incorporating environmental variations in a microcosm study. The oil biodegradation extent was determined by measuring the remaining oil and grease in the sample vials. The highest percentage of biodegradation was 88.6%, with a constant degradation rate of 0.399 day^–1^. GC-MS analysis showed that the most degradable compound in the oil samples was paraffin. This study also observed that microbial degradation was optimized within three days of exposure and that degradation ability decreased at 35 °C. The salinity variation effects were insignificant. Based on all analyses, deep-sea sediment bacteria have great potential in oil spill biodegradation in a microcosm scale.

## 1. Introduction

Oil spills are a severe cause of marine environmental pollution that result from intentional or accidental activities such as tanker accidents during marine oil transportation, offshore oil exploration and exploitation, and oil and fuel run-off from the land [1]. Once spilled into the marine environment, petroleum hydrocarbons and their products are subject to various natural processes such as evaporation, emulsification, dispersion, photo-oxidation, biodegradation, and sedimentation [2]. In many cases, oil accumulation in an environment occurs faster than its recovery from a spill. Consequently, applying appropriate technology is necessary to remediate oil spill pollution [3].

Although much of an oil spill can easily be dispersed in the water column, a portion may also settle in deep-sea sediments [4]. This happens through adsorption to suspended solids that subsequently sink into the underlying sediments in many cases. Sedimentation of spilled oil also occurs when oil reacts with oil dispersants to produce denser oil. The dispersed oil can then be deposited into bottom sediments via oil-plume settling [5].

Most previous work on oil spill biodegradation has been conducted using hydrocarbon-degrading bacteria from the water column and coastal sediments [6,7,8,9]. Some studies also suggest that deep-sea sediment bacteria have been recognized as capable of hydrocarbon degradation [10]. In deep-sea sediment, bacterial communities respond to hydrocarbon pollution through aerobic processes in surface sediments and anaerobic processes in subsurface sediments.

This study examined the potential use of deep-sea sediment bacteria for the biodegradation of oil spills with environmental variations on a laboratory scale. The experiment was performed using mixed culture bacteria from three dominant bacterial isolates that have the highest population in deep-sea sediments. This study also investigated the biodegradability of crude oil, both aliphatic and aromatic compounds, under various experimental conditions, as a function of contact time between bacteria and oil, temperature, and salinity. Changes in oil content were analyzed using a GC-MS instrument before and after the biodegradation treatment.

## 2. Materials and Methods

### 2.1. Experimental Design

The experiment was carried out in four microcosm scales, i.e., three microcosms as replicates and one as the control. To each microcosm were added 3000 ppm crude oil, no nutrient, and three species in consortium bacteria isolated from sediment samples. No addition of consortium bacteria was added to the control microcosm. The design consisted of bacterial isolation from deep-sea sediment samples, cultivation, identification, bacterial adaptation to an oily environment, biodegradation processes, and variation to environmental change (Figure 1).

The initial condition of crude oil in this experiment was selected according to the crude oil properties before biodegradation. Percentage change in crude oil, as an indicator of oil-degradation bacteria, was measured in each of seven solution mixtures taken from seven sediment sample sites after 7 days of incubation.

### 2.2. Bacteria Isolation, Cultivation, and Identification

The deep-sea bacteria used were provided by the Biotechnology Laboratory of Center for Research and Development of Technology for Oil and Gas (LEMIGAS). Deep-sea sediment samples were collected using gravity cores from seven locations at more than 1000 m of depth in Indonesian Makassar Strait and the Flores Sea (Figure 2). All core samples were kept in sterile conditions at a temperature of 4 °C during transportation and storage in the laboratory.

The bacteria were aseptically isolated from deep-sea sediment based on the methods developed by previous research [8,11]. The suspension was prepared by adding 1 g of freeze-dried deep-sea sediment samples for each location to a sterile test tube, and then homogenized with sterile sea water using a vortex shaker. A total of 1 mL of the inoculum-containing water was pipetted and transferred into a test tube containing 9 mL physiological solution (NaCl 9%) and mixed. The mixture was consecutively diluted three times with a ratio of 1:9 for each dilution tube. A total of 0.1 mL of each mixture tube was pipetted and transferred into petri dish containing 10–15 mL plate count agar (PCA). The use of PCA as a bacterial growth medium aimed to calculate the total number of bacteria in deep-sea sediments. The culture was incubated for seven days, and all of these processes were conducted in a laminar air flow bench to avoid contamination. The results of bacterial growth were processed and selected to obtain the three bacterial isolates with the highest abundance and these were used in the experiment.

Identification of the isolated bacteria was based on the Biolog Gen III identification system (Biolog Inc., Hayward, CA, USA). The bacterial identification used in this study was a physiological profiling method of Biolog Gen III according to the manufacturer’s instructions. A single colony was selected and emulsified into inoculation fluid as a suspension. A total of 100 μL of the cell suspension was inoculated into a GN2 MicroPlate test plate and incubated for 16–24 h at 37 °C. GN2 MicroPlates were read in a MicroStation Reader. The identification process was conducted using a pre-loaded ID database GEN III database, version 5.2.1 (Biolog Inc., Hayward, CA, USA) and interpreted by the identification system’s software by identifying species that had close reaction patterns along with the percentage of similarity. The identified bacteria were then cultivated in 100 mL of nutrient broth in the flask.

Selected bacterial isolates that had been cultivated were then adapted periodically every 72 h. A total of 10 mL of cultivated bacteria was pipetted and transferred into 100 mL of conditioned seawater media containing 0.3% (*v*/*v*) crude oil. The crude oil (mid-heavy crude, 34.48° API, density of 0.8474 g cm^−3^) was used as an adaptation media for an oily environment. The conditioned seawater was prepared by dissolving 1.26 g of MgSO_4_·7H_2_O, 1 g of KCl, 2.5 g of KH_2_PO_4_, 3.75 g of Na_2_HPO_4_, and 1.29 g of NaNO_3_ into 3 L of sterile seawater. The mixed solution was then shaken at room temperature using a reciprocal shaker at 120 rpm [12]. Previous studies show that these mixed culture bacteria have adapted successfully to their oily environments [11].

### 2.3. Biodegradation Process

#### 2.3.1. Time-Series Biodegradation Experiment

The three isolates from each station were adapted as mixed culture bacteria and then pipetted 10 mL into sterile sea water along with 0.531 mL crude oil equivalent to 3000 ppm in each microcosm. The mixture of bacteria and oil was put into an environmental shaker and shaken at room temperature at 120 rpm. Biodegradation processes were observed at days 0 and 7 by measuring the oil and grease content, pH, and bacterial population.

The determination of oil and grease used the gravimetric principle referring to ASTM D-4281-95 [13]. Samples were poured into a separating funnel, added to 60 mL of *n*-hexane, and shaken for 10 min. The upper layer extract (a mixture of oil and hexane) was poured into a pre-weight boiling flask. The extract was then evaporated using a rotary evaporator at 70–80 °C. After cooling, the boiling flask was weighed. The rate of biodegradation is calculated by the following equation [8]:(1)Ct=Co exp(−kt),
where Ct is oil concentration over time t (mg L^−1^), Co is the initial oil concentration on day 0 (mg L^−1^), k is biodegradation rate coefficient (day^−1^), and t is time (day).

#### 2.3.2. pH and Bacterial Growth

The pH of the mixture was determined using a standard pH meter, Boeco BT-600, calibrated with pH 7 and pH 4 buffer solutions. The mixed bacterial populations were measured using an optical density method with a Boeco S-22 UV/V is spectrophotometer at 600 nm. An absorbance of 0.7 was equivalent to a population of 10^8^ CFU mL^−1^ [14]. The bacterial population growth rate was calculated using the following equation [15]:(2)$$\mu =(\ln N_{t}-\ln N_{0})/t,$$
where µ is the rate of growth of the population abundance of total bacteria (day^−1^), N_0_ is the initial abundance of the bacterial population (CFU mL^−1^), N_t_ is the abundance of the bacterial population at the t observation of biodegradation (CFU mL^−1^), and t is the time interval from N_0_ to N_t_ (day). The measurement of oil and grease, pH, and bacterial population was repeated three times.

#### 2.3.3. Environmental Variations

Environmental conditions such as salinity and temperature for the growth of bacteria and their survival were studied during oil biodegradation. Those parameters were monitored on days 1, 2, 3, 4, and 7 [12]. The variations were chosen according to the nature of the seawaters (Makassar Strait and the Flores Sea, Indonesia), i.e., ranging from 30.5–34.5 psu [16] upper limit and 20 psu coastal ranges as the lower limit, and sea surface temperature conditions: 25–34 °C [17]. Therefore, the salinity treatments used during the experiment were 20, 25, 30, and 35 psu, and temperature treatments were 28, 32, and 35 °C.

### 2.4. Crude Oil Analysis

#### 2.4.1. Oil Fractionation

Oil samples with the highest percentage of degradation (Station 18544) were then analyzed using GC-MS to see the maximum potential of degrading bacteria in this microcosm simulation. Degraded crude oil was fractionated using a column chromatography method to separate the aliphatic and aromatic compounds in the oil. Fractionation was performed by eluting neutral fractions with 150 mL of *n*-hexane solvent on a column filled with silica gel to obtain the aliphatic fraction. Aromatic fractions were eluted by obtaining the neutral fraction using 50 mL of *n*-hexane and dichloromethane (1:1). Each fraction was evaporated (without nitrogen) to obtain approximately 1–2 mL, and the sample was transferred to a vial. The fraction in the vial was then evaporated with nitrogen until it was dry. The 0.5 mL *n*-hexane solvent was added to the vial prior to gas chromatography–mass spectrometry (GC-MS) analysis [18].

#### 2.4.2. GC-MS Analysis

Oil fractionation results were analyzed using an Agilent GC-MS System 7890A with an Agilent Inert MSD 5975C detector, HP-5MS Ultra Inert GC column (60 m × 0.250 mm × 0.25 μm), and ultra-high pure (UHP) helium as the carrier gas. This method defined the composition and types of compounds contained in the sample before and after biodegradation. The gas chromatography oven was programmed at 16.08 kPa with a column flow of 1 mL min^−1^. The temperature began at 40 °C and rose to 250 °C; the samples were run at a rate of 10 °C min^−1^ and then held for 25 min. The temperature was increased again from 250 °C to 290 °C at a rate of 10 °C min^−1^ and then held for 30 min. The sample of oil used for GC-MS analysis was taken from the station with the highest percentage (Station 18544) of oil biodegradation. The unit of abundance in the results refers to total hydrocarbons.

Further analysis of chromatogram was carried out by calculating the abundance of pristane (Pr) and phytane (Ph), which are commonly used as markers to observe the biodegradation process in crude oil. The comparison of *n*-C_17_/Pr and *n*-C_18_/Ph abundance was carried out to ensure that the degradation process occurred biologically, and the carbon preference index (CPI) was analyzed to show the ability of bacteria to degrade odd-numbered and even-numbered alkanes at the same ratio.

### 2.5. Statistical Analysis

The effects of treatments on the oil degradation were tested using an analysis of variance as well as using the least significant difference (LSD) and Duncan multiple range tests for further analysis. This analysis also assessed the rate of oil degradation and its relation to other parameters using simple regression and correlational analysis.

## 3. Results

### 3.1. Analysis of Crude Oil Biodegradation

The consortium of three bacteria (*Raoultella* sp., *Enterobacter* sp., and *Pseudomonas* sp.) was used for the crude oil biodegradation process. The analysis of crude oil using isolated bacteria from different stations shows different percentages of the oil biodegradation process. The percentages varied from 59.55% (Station 18523) to 88.64% (Station 18544) (Figure 3) and the biodegradation process positively correlated with the size of the bacterial population, whereas the opposite occurred to the pH of the mixed solution.

### 3.2. GC-MS Analysis of the Crude Oil Components

#### 3.2.1. Paraffin

A carbon number series of *n*-C_11_–*n*C_27_ aliphatic compounds were identified in each sample observed before and after the biodegradation test (Figure 4). However, their intensities were different due to the decreasing factors of bacterial biodegradation. A significant decrease occurred in hydrocarbons with low molecular weights, such as *n*-heptadecane (*n*-C_17_) and *n*-octadecane (*n*-C_18_).

The ratio of Pr/Ph was relatively unchanged before and after biodegradation (Table 1); this was likely due to their similar degradation rates. The ratios of *n-*C_17_/Pr and *n-*C_18_/Ph estimate how bacteria can selectively degrade the hydrocarbons in crude oil. After the biodegradation test, the decrease in *n*-C_17_/Pr and *n*-C_18_/Ph ratios shows that degradation proceeded successfully with reduction ratios of 0.32 in *n-*C_17_/Pr and 0.50 in *n-*C_18_/Ph. The CPI values of C_11–19_ and C_20–27_ after the test were also unchanged.

#### 3.2.2. Aromatics

Aromatic compounds, consisting of carbon numbers C_7_–C_14,_ had diverse abundance. The results also indicate bacterial degradation, corresponding to the ratio of the abundance of aromatic compounds after biodegradation. Some aromatic compounds were completely degraded, whereas others were only slightly degraded (Figure 5).

The BTX compounds were not detected in this analysis, presumably because the headspace of the GC-MS was not used during the analysis, in addition to the already low concentration of BTX in crude oil after the biodegradation process. On the other hand, naphthalene, anthracene, and phenanthrene were slightly degraded. The identified naphthenic compounds (Figure 5) show various abundance levels in carbon numbers C_9_–C_10_. GC-MS analyses identified naphthenic compounds and the results show that there was a degradation. The abundance of naphthenic compounds was small compared with the paraffinic and aromatic compounds (Figure 6).

The paraffinic compounds were readily biodegraded compared with the aromatic and naphthenic compounds. The paraffinic compounds decreased from 36.52% to 12.58%, whereas the aromatics decreased from 20.72% to 10.96%, and the naphthenic compounds from 3.50% to 0.23%.

### 3.3. Degradation Rate and Environmental Variations

Time variation aimed to examine the rate of oil degradation in bacteria with the best ability to degrade the oil. The decline of the crude oil concentration was observed during the experiment (Figure 7a). Initially, the oil and grease concentration was measured at 2944.5 mg L^−1^, followed by a significant decrease on day 1 to 1665.5 mg L^−1^ (or degradation of 43.4%). On day 2, the oil and grease content decreased to 1064 mg L^−1^ (or by 63.9%). On day 3, the oil and grease content decreased to 407.6 mg L^−1^ (or by 86.5%). The degradation rate was 0.399 day^−1^.

Although the degradation levels appeared different on day 4 and day 7, they were insignificant compared with that on day 3. Oil and grease content on day 4 and day 7 were 397 mg L^−1^ and 346.5 mg L^−1^ (or they were degraded by 86.5% and 88.2%), respectively.

Statistical analysis (analysis of variance) was used to determine the effect of time variation on days 1, 2, 3, 4, and 7. The duration of contact between the bacteria and the oil had a significant effect on biodegradation (Sig. on LSD is 0.00; *p* < 0.01). LSD and Duncan tests show a significant decrease in oil content from day 1 to day 3 and located on different subsets on the Duncan test. However, the tests on days 4 and 7 show similarities with day 3, as seen from the Sig. on LSD is >0.01 and located on the same subset in the Duncan test. The time variation effect suggests that day 3 was the optimum time for biodegradation of the oil. Changes in bacteria population and pH occurred as well (Figure 7b).

Table 2 shows that the bacterial growth rate increased continuously from day 0 to day 7. The bacterial population increased, and the maximum growth rate was μ = 0.10 on the third day of the biodegradation process, which corresponded to the fact that the oil concentration decreased significantly on day 3. Afterwards, there were insignificant differences in the abundance of bacteria. This is shown by the decreased growth rate (µ) on day 4 and day 7. In this study, since the mixed bacteria degraded the oil most optimally on day 3, the environmental variations were applied only until then.

The declining percentage of the oil content on salinity variation ranged between 86–87% (Figure 8a). The temperature used in the salinity variation was at a controlled condition of 28 °C. Figure 8a indicates the bacteria were able to grow in various saline conditions. The oil degradation declined as the temperature increased (Figure 8b), and according to the LSD and Duncan tests, the significant effect occurred at 35 °C, so that the biodegradation process was found to be effective and efficient at temperatures below 35 °C.

## 4. Discussion

Although oil spills can naturally decrease due to several processes such as evaporation, emulsification, and dispersion, some may be deposited in the bottom of the seabed. Other research found that hydrocarbons reached deep-sea sediment following tragedies such as the Montara oil spill, Indonesia, and Deepwater Horizon, Gulf of Mexico [19]. The occurrence of deep-water bacteria is expected to further degrade the remains of the oil in deep water. This study shows that deep-sea bacteria tested in a closed system could degrade oil within 3 days, varying between 59.55% and 88.64% degradation (Figure 3a). These isolated bacterial mixtures consumed crude oil hydrocarbons, causing a decrease in the concentration of crude oil in the mixed medium [9]. The process of biodegradation in this experiment is indicated by the increase of the bacterial population and the reduction of both pH level and oil content (Figure 2). The increase of the bacterial population indicates that the bacteria grew by utilizing oil as an energy and carbon source in their metabolism.

In addition to the growth of the bacteria, the carbon in the oil decomposed aerobically and was followed by a decrease in pH due to carbon dioxide production [20]. The bacteria in this study were able to grow under the experimental conditions because deep-sea sediment bacteria can live in a pH range of 5 to 8 [21]. Moreover, the patterns of these three parameters showed similar results after 3 days of the incubation process. The biodegradation of the oil only occurred until the third day, as afterwards the consortium apparently entered the stationary phase and stopped growing. The decrease in the degradation rate at the following stage could possibly be due to nutrient depletion, such as nitrogen (N) and phosphorus (P) [22]. In this research no additional nutrients were given and carbon was assumed to be the sole food source for the bacteria as the experiment was carried out in a limited and closed system on a controlled microcosm scale. Biodegradation by bacterial metabolism produces final products such as organic acids and carbon dioxide [20], leading to a lower pH because carbon dioxide reacts with water to form acidic acid H_2_CO_3_; further acidification is due to the presence of organic acids.

Biodegradation occurred in paraffinic, aromatic, and naphthenic compounds (Figure 6), but the most commonly degraded components were aliphatic hydrocarbons (Appendix A). The complexity of oil biodegradation depends on the structure and molecular weights of the compounds. Previous research revealed that hydrocarbon compounds with high molecular weights were more challenging to degrade due to their low solubility [23]. Many researchers reported that aliphatic compounds could be degraded easily since they have a long-chain carbon bond that the bacteria can break through alpha and beta-oxidation. A significant decrease occurred in these compounds with low molecular weights, as shown by the GC-MS analysis results, such as *n*-heptadecane (*n*-C_17_) and *n*-octadecane (*n*-C_18_). Selective biodegradation can also be assessed by examining the ratio of *n*-heptadecane to isoprenoid pristane (*n*-C_17_/Pr) and n-octadecane to isoprenoid phytane (*n*-C_18_/Ph) [24]. In addition, other research [12] revealed that other indicators of biodegradation were the ratios of *n*-C_17_/Pr and *n*-C_18_/Ph; this experiment shows that these ratios decreased after three days of incubation. The carbon preference index (CPI) represents linear odd–even carbon alkanes [18]. This numerical calculation shows the dominance of odd or even alkanes in a particular carbon number range, where CPI values > 1 indicate a predominance of odd carbon alkanes. In this result (Table 1), bacteria could degrade odd- and even-numbered alkane compounds in the same ratio.

Bacteria *Raoultella* sp. in the mixed culture can degrade aromatic compounds. These compounds include monocyclic aromatic hydrocarbons, which significantly affect water pollution, although only in small quantities [25]. They are divided into two groups, BTX compounds, which were under detection limit after biodegradation, and naphthenic compounds, which were only slightly degraded by the bacteria (Figure 5). The most difficult compounds to degrade included those with two or more aromatic rings within the chains, such as naphthalene, anthracene, and phenanthrene (Appendix A). Aromatic compounds with two rings and more have tremendous resonance energy; hence, more energy is required to break the chainrings of these compounds. Toluene and xylene (1,2-dimethylbenzene) have a methyl group, enabling benzene rings to create electrophilic substitutions. As a result, toluene and xylene are more reactive and are easier to degrade by bacteria than naphthalene.

Although found in small amounts, naphthenic compounds are among the most toxic components of crude oil [26]. Biodegradation of naphthenic compounds begins with methyl substitutions on the cycloalkane rings by the bacteria consortium [27]. One of the bacteria used in this study (*Pseudomonas* sp.) is considered one of the few known naphthenic-degrading microorganisms [28].

The paraffinic, aromatic, and naphthenic compounds dominated all of the compounds found in oil with a C_7_ to C_27_ carbon distribution. After biodegradation over three days, there was a notable decline in these compounds. According to previous research, the oil biodegradation process is characterized by differences in the concentrations of each carbon chain to a compound constituent of crude oil before and after the treatment [12]. Figure 6 shows a decrease in the abundance of paraffinic, aromatic, and naphthenic compounds in the samples.

In this study, there are indications of environmental influences, especially temperature, where the biodegradation process at 35 °C was less effective. High temperature (such as 35 °C) causes proteins, nucleic acids and other cellular components of bacteria to be broken down, which results in disruption of the biodegradation process. These bacteria are capable of degrading oil at an optimal range of temperature between 28–31 °C [21]. If the temperature is above 31 °C, then the rate of degradation decreases rapidly [20]. This study shows that at a temperature of 35 °C, bacteria could only degrade oil as much as 66.65%.

Based on the statistical analysis using LSD and Duncan tests, salinity variation did not significantly affect the biodegradation process. The deep-sea sediment bacteria were able to degrade crude oil under various saline conditions. Therefore, these bacteria may be suitable bioremediation agents for oil spills in deep waters. [29,30].

In general, this study indicates that deep-sea sediment bacteria are capable of degrading oil spills based on laboratory experiments and have the potential to be used in the remediation of oil spills in aquatic environments. Previous results from other researchers are consistent with this research using the same genus of bacteria: *Raoultella, Enterobacter,* and *Pseudomonas* [12,23,31,32,33]. Further research should be carried out to find specific species that can degrade hydrocarbons using other types of media specifically for hydrocarbon-degrading bacteria, e.g., Bushnell–Haas or ONR7, on a larger scale.

## 5. Conclusions

Deep-sea sediment bacteria have the potential to remove oil spills on a microcosm scale. The highest percentage of biodegradation by a mixed culture of deep-sea sediment bacteria was 88.6%, with a constant degradation rate of 0.399 day^−1^. GC-MS analysis shows paraffin is the most degraded compound. Environmental conditions, especially temperature levels, are crucial for the biodegradation process.

## Figures and Tables

**Figure 1 microorganisms-10-01616-f001:**
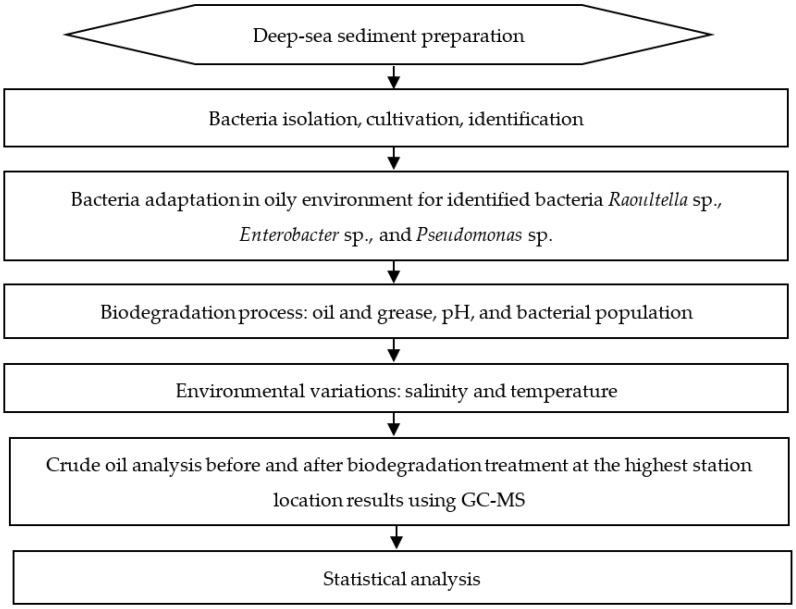
Schematic diagram of oil biodegradation research using deep-sea sediment bacteria.

**Figure 2 microorganisms-10-01616-f002:**
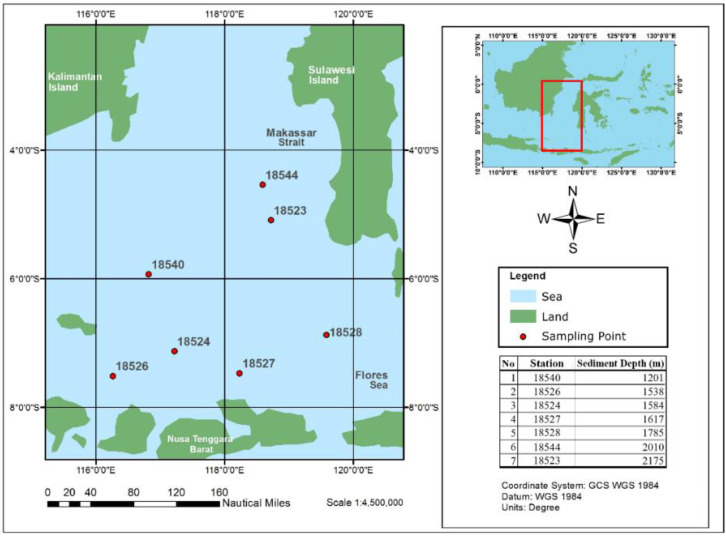
Seabed location of the deep-sea sediment sampling in the surrounding Makassar strait.

**Figure 3 microorganisms-10-01616-f003:**
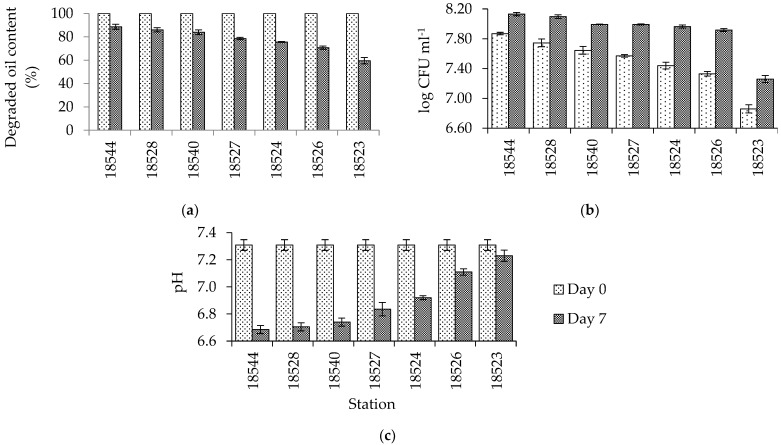
The properties of the oil degradation experiment with consortium bacteria extracted from 7 sediment samples before and after a 7−day experiment: (**a**) percentage of oil and grease content, (**b**) the number of the mixed bacterial population, and (**c**) pH condition with standard deviation.

**Figure 4 microorganisms-10-01616-f004:**
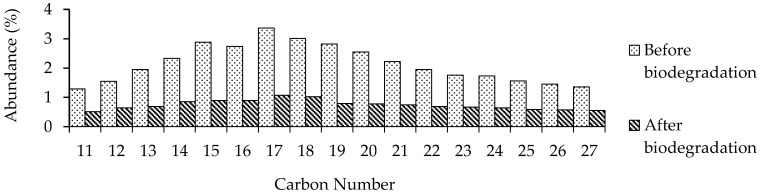
Content change of paraffinic compounds found in crude oil before and after biodegradation from GC-MS analysis. The abundance of linear alkanes with chain lengths between 11 and 27 carbon atoms in the oil phase before and after the biodegradation process.

**Figure 5 microorganisms-10-01616-f005:**
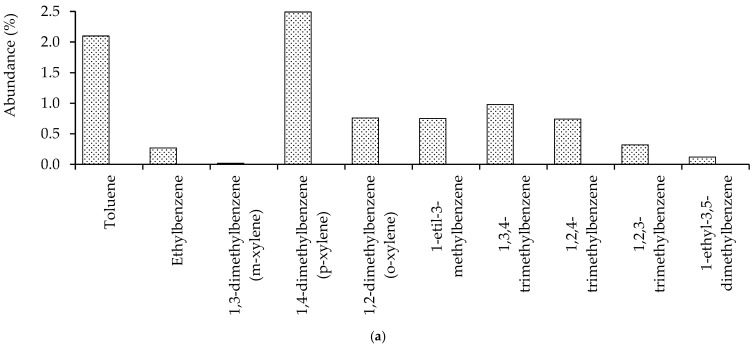
Content change of (**a**) aromatic compounds and (**b**) naphthenic compounds found in crude oil before and after biodegradation from GC-MS analysis. Benzene, toluene, and xylene (BTX compounds) are under detection limit after biodegradation.

**Figure 6 microorganisms-10-01616-f006:**
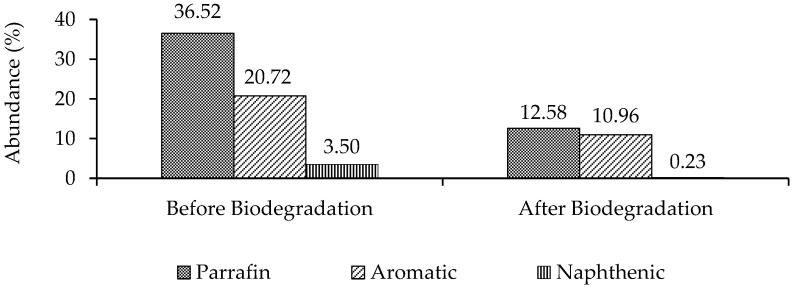
GC-MS showed the abundance of paraffinic, aromatic, and naphthenic compounds in the oil samples. Those entire components were degraded by the biodegradation process.

**Figure 7 microorganisms-10-01616-f007:**
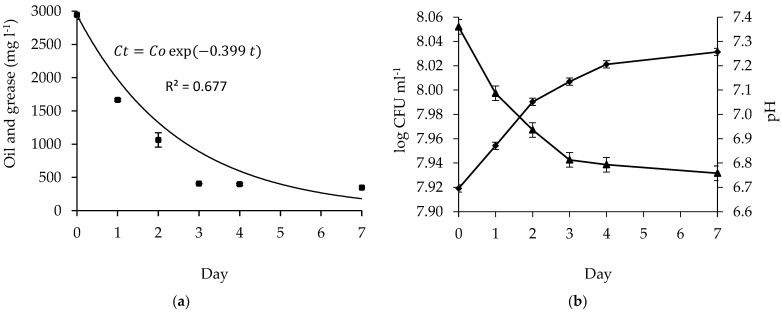
(**a**) The biodegradation rate of oil (
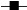
), (**b**) bacteria population (
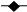
), and pH (
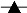
) during experiment.

**Figure 8 microorganisms-10-01616-f008:**
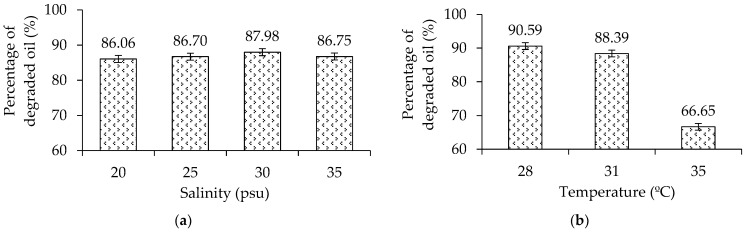
Percentage of degraded oil on (**a**) salinity variation at controlled temperature 28 °C and (**b**) temperature variation after 3 days of incubation. At 35 °C, the oil degradation process by bacteria was ineffective.

**Table 1 microorganisms-10-01616-t001:** Pr/Ph ratio, *n-*C_17_/Pr, *n*-C_18_/Ph, and carbon preference index (CPI) on crude oil before and after biodegradation.

Parameter	Before Biodegradation	After Biodegradation
Pr/Ph	2.25	2.25
*n-*C_17_/Pr	0.84 ^1^	0.52
*n*-C_18_/Ph	1.54 ^1^	1.04
CPI C_11–19_	0.99	0.99
CPI C_20–27_	0.96	0.96

^1^ The crude oil may have weathered naturally, because usually fresh crude oil has a value of 1.4–2.5 for *n-*C_17_/Pr and 2.5–15 for *n*-C_18_/Ph. However, this research focused on how biodegradation works.

**Table 2 microorganisms-10-01616-t002:** Population and specific growth rate of bacteria on biodegradation process.

Day	Bacteria Population (CFU mL^−1^)	Growth Rate, µ (day^−1^)
0	8.3 × 10^7^	-
1	8.5 × 10^7^	0.02
2	9.2 × 10^7^	0.08
3	1.0 × 10^8^	0.10
4	1.1 × 10^8^	0.08
7	1.2 × 10^8^	0.03

## Data Availability

Not applicable.

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
