# Peer review of "Potential Use of Deep-Sea Sediment Bacteria for Oil Spill Biodegradation: A Laboratory Simulation"

_microorganisms, 2022, doi:10.3390/microorganisms10081616_

Round 1

Reviewer 1 Report

The paper reports on experiments showing that bacterial isolates from sediment samples in the Flores Sea, off the coast of Indonesia, can degrade petroleum hydrocarbons in microcosms. It is an important and reassuring observation, to know that oil spills reaching deep-sea sediments, can be degraded by indigenous communities. The effects of various conditions (temperature, salinity) have been investigated and the oil profiles were characterized by GC-MS.

I am not sure on the issue of “bioremediation”. I don’t think it makes sense to dredge polluted sediments from 1 km depth, bring them to the surface and clean them using bacterial cultures. It is much more useful to let the pollution in place and allow for in situ degradation (if we are sure it happens). So while the experiments are OK, I am not sure whether the authors have the right goal in mind.

An issue missing in this paper is a full characterization of the bacterial isolates. The isolates were typed only using physiological profiling (Biolog). Nowadays a molecular characterization is easy to do (!6S rRNA screen). This is especially relevant for a microbiological journal, but it could be of lesser importance in a journal like Biodegradation.

The manuscript requires English editing; e.g. the title is already very cumbersome, it could read “Oil spill degradation by deep-sea bacterial isolates”.

Author Response

Dear Reviewer 1,

First of all, we would like to thank you for your constructive review and suggestions. All suggestions and comments were helpful in our revision process. I am sending you the author’s response and revised manuscript as well (attached).

On behalf of all authors, I express my gratitude and many thanks to the reviewer and editor, who have done a very great job for this one of an important part of the publication process.

I am looking forward to receiving your response.

Yours sincerely,

Tri Prartono, Angga Dwinovantyo, Syafrizal Syafizal, and Agung Dhamar Syakti

Reviewer 2 Report

The authors describe the isolation and phenotypical identification of bacteria from three different Proteobacteial genera from Deep Sea sediment samples. Subsequently, they investigated the performance of consortia consisting of the Deep Sea-derived Pseudomonas, Enterobacter and Raoultella with regard to the degradation of crude oil under different conditions.

The overall study is valid but some points may be better explained:

1. Why were PCA and NB used as media for isolation instead of Bushnell-Haas or ONR7 or something similar established to isolate specific hydrocarbon degraders? The choice of media would probably exclude the obligate and highly effective hydrocarbon degrading phyla like Cycloclasticus or Alcanivorax that are unable to utilize trypton and glucose for growth.

1b Of course, it is a valid approach to enriching for oil-degrading Enterobacteria or Pseudomonas but I think it should be mentioned as a specific target of study, maybe even specified  already in the title.

2. Mere phenotypical identification of species without support by molecular techniques like (partial) 16SrRNA sequencing appears a bit uncommon from a microbiological point of view but may be valid in the field of application although it will probably make it impossible for other groups to reproduce and compare the results in detail. However, without any molecular support, the author cannot state, that they isolated the same strains from all the locations as is done in line 185f as the databases list hundreds of species and strains of this genera.

Even the results presented in FIg. 3 indicate that the strains differ as achieved cell densities and acidification of the medium differ between the consortia isolated from the different sites.

3. Given this, an explanation is missing which consortium was used to conduct the subsequent experiments? Was the enterobacter/Pseudomonas/Raoultella mix from one specific station or a mix of all of them?

4. Likewise stated should be the temperature during the experiments in Fig 3-8a. Is it 28°C like the lowest temperature in Fig. 8b? Or 25° or 20°C? If a lower temperature was applied than the one investigated in Fig. 8, it should be clearly stated how much oil was degraded in these experiments to be compared with the information given in Fig8b.  (I assume something like 86% as indicated in 8a?)

5. For a clearer presentation of the results, I suggest combining the graphs of Figure 4, 6 and 7 into one figure and putting the mass spec raw data into a supplementary file.

6. It would be nice if the authors can also define the term abundance in this context just like as they provided for degradation and growth rate.

7. Regarding figure 7, the authors may consider that biodegradation of the oil only occurs until day 3 as afterwards the consortium apparently enters the stationary phase and stops growing maybe because of the depletion of carbon or nitrogen sources (Fig 7b).

7b: It would be interesting to know if changes in the composition of the bacterial community occurred during the degradation? Did one of the strain grew first and than the others? Or was one species enriched while outgrowing the others? This is not mandatory to make the manuscript worth publishing but would help to gain impact because specific contributions of the "microbial partners" could be deduced.

8. As the study aimed to assess Deep-Sea sedimental  enterobacterial communities for oil degradation compared to water column and surface water communities, it should be discussed how the studies community performed in comparison to the mentioned studies with water column bacteria.

9. The language is sometimes hard to follow and occasionally even contradictive ("The experiment was carried out on a microcosm scale. The experiment was carried out in four mesocosm scales" ). It should be tried to be improved towards enhanced clearness e.g. in the figure captions, e.g." The abundance of linear alkanes with chain lengths between 11 and 27 carbon atoms in the oil phase before and after the biodegradation process" instead of "The abundance of carbon numbers 11 to 27 in crude oil decreased after biodegradation process" (l. 208).

Author Response

Dear Reviewer 2,

First of all, I would like to thank you for your constructive review and suggestion. All suggestions and comments were helpful in our revision process, and have been adopted in the revised paper that had been submitted. We have tried to revise the English used in this manuscript with the help of native speaker for the grammatical check. The manuscript was revised based on your suggestion as well, especially in the Method section. I am sending you the author’s response (attached).

On behalf of all authors, I express my gratitude and many thanks to the reviewers and editor who have done the very great job for this one of an important part of the publication process.

I am looking forward to receiving your response.

Yours sincerely,

Tri Prartono, Angga Dwinovantyo, Syafrizal Syafizal, and Agung Dhamar Syakti

Reviewer 3 Report

This is a well-established paper to study the spilled oil degradation rates by three deep-sea bacteria. I support its publication. However, the authors have to fix some major flaws in this manuscript, especially in the method section, which is too general and hard to repeat in other labs. Here are some specific comments that need to be addressed.

1. Line 58 -64. what is the water that is used for the mesocosm scale study? Is there any specific pretreatment applied? Where is the crude oil come from? Could you list some basic properties of the crude oil and the mesocosm system after mixing with the oil?

2. Delete Line 88-93 because the authors wrote them again later in this manuscript.

3. Line 106-107. What's the volume of "inoculum containing water" used? How did the authors prepare the agar plate? Needs details here.

4. Line 110. Please add details of the LB broth.

5. Line 110-112. Could you provide some details of the bacteria identification?

6. Line 133. It's unclear what the meaning of "t is time in a day" is?

7. Line 140. the unit of growth rate is 1/day according to equation (1).

8. Line 195. Please add the standard deviation after the average value, which applies to the rest of the results and discussion section.

9. Table 1. Where are those data from? Are that measured data? I didn't see the authors discuss that in the method section. I apologize if I missed it. It would be good if the authors could add some comments on each parameter in the previous section. 

10. Figure 5. I suggest the authors use "under detection limit" instead of "disappeared."

11. Figure 6. Where are the error bars in this figure?

Author Response

Dear Reviewer 3,

First of all, I would like to thank you for your constructive review and suggestion. All suggestions and comments were helpful in our revision process, and have been adopted in the revised paper that had been submitted. We have tried to revise the English used in this manuscript with the help of native speaker for the grammatical check. The manuscript was revised based on your suggestion as well, especially in the Method section. I am sending you the author’s response (attached).

On behalf of all authors, I express my gratitude and many thanks to the reviewers and editor who have done the very great job for this one of an important part of the publication process.

I am looking forward to receiving your response.

Yours sincerely,

Tri Prartono, Angga Dwinovantyo, Syafrizal Syafizal, and Agung Dhamar Syakti

Round 2

Reviewer 1 Report

A few editorial issues remain:

Line 24: 0.339 – what are the units here. Is it a rate per day or per week?

Line 36: product = products

Line 65: Each microcosm condition was added = To each microcosm were added

Line 157: the degradation rate constant k has unit day–1.

Line 164: the unit of µ (growth rate of the bacterial population) is day–1, not CFU ml–1 day–1.

Line 219: attained through matrix-assisted … – I don’t understand what you mean here. The consortium was characterized by Biolog, not by MALDI MS. There is no mention of MALDI MS in the Methods section. Please correct.

Line 293 – unit required here.

Line 309, Table 2: the unit of µ (growth rate of the bacterial population) is day–1, not CFU ml–1 day–1.

Line 380: “These compounds are part of a monocyclic aromatic compound” = “These compounds include monocyclic aromatic hydrocarbons”

Line 387: was required = is required

Line 388: “have a -CH3 alkyl group” = “have a methyl group”

Line 399: “suggesting that … “ – propose to skip this line, since it does not follow logically from the previous sentence.

Author Response

Dear Reviewer 1,

We have studied your comments carefully and have made revision which marked in the paper. Attached please find the revised version, which we would like to submit for your kind consideration.

On behalf of all authors, we thank you very much for giving us an opportunity to revise our manuscript, we appreciate editor and reviewers very much for their positive and constructive comments and suggestions on our manuscript.

We really hope that you accept this revised version so that the manuscript will be accepted and published. Once again, thank you very much.

Yours sincerely,

Tri Prartono, Angga Dwinovantyo, Syafrizal Syafizal, and Agung Dhamar Syakti

Reviewer 2 Report

Thanks to the authors for improving the manuscript.

Still, some minor questions remain:

Regarding question 2 :

[...] The scale used was also microcosm, thus from our test results it can be concluded that the isolated bacteria allowed for biodegradation.

Thank you for the clarification.  That is right, this conclusion can be drawn from the experimental data and that's all fine. However, with the phenotypic taxonomy approach allowing only the determination of the Genus level but not the species not to mention a specific strain, the statement "The bacterial strains used in this study were found in samples from all stations" (l.221) cannot be proven, as the databases list hundreds of species and strains of these genera, and should be removed.

For the same reason, I asked to specify from which station the consortium was sampled that was used for the detailed degradation study. The methods section now mentions "the three isolates" but in my opinion, it cannot be excluded that up to 21 isolates were obtained with each sampling spot being inhabited by a distinct consortium of Pseudomonas, Enterobacter and Raoultella. If analyses of the bacteria obtained from station 18544 are shown in 3.2 and 3.3, fine, it appears perfectly reasonable to focus on that community given the presented data, but it just should be written down.

Regarding remark 6, Thank you, again, for providing definitions for "abundance". This is really helpful. I already thought that "the abundance of degradation refers to the total composition of hydrocarbons. " was meant but I wondered why the abundance of every compound goes down. If "abundance" means the relative amount of the named compound in total hydrocarbon there have to be compounds that are enriched (e.g. in Fig 37% Parrafin, 21%Aromatic 4% naphthenic, 39% other stuff are degraded to a mixture of 13%, 11%, and 0.2%, respectively and  76 % other stuff. That is impressive but did the experiments yield any idea which components of the hydrocarbon phase are the "other stuff" that is not degraded?

I apologize if I misunderstand the study on these points but from my current point of view, these questions are not solved.

Author Response

Dear Reviewer 2,

We have studied your comments carefully and have made revision which marked in the paper. Attached please find the revised version, which we would like to submit for your kind consideration.

On behalf of all authors, we thank you very much for giving us an opportunity to revise our manuscript, we appreciate editor and reviewers very much for their positive and constructive comments and suggestions on our manuscript.

We really hope that you accept this revised version so that the manuscript will be accepted and published. Once again, thank you very much.

Yours sincerely,

Tri Prartono, Angga Dwinovantyo, Syafrizal Syafizal, and Agung Dhamar Syakti
